# The Role of Polymeric Coatings for a Safe-by-Design Development of Biomedical Gold Nanoparticles Assessed in Zebrafish Embryo

**DOI:** 10.3390/nano11041004

**Published:** 2021-04-14

**Authors:** Pamela Floris, Stefania Garbujo, Gabriele Rolla, Marco Giustra, Lucia Salvioni, Tiziano Catelani, Miriam Colombo, Paride Mantecca, Luisa Fiandra

**Affiliations:** 1POLARIS Research Centre, Department of Earth and Environmental Sciences, University of Milano-Bicocca, 20126 Milan, Italy; pamela.floris@unimib.it; 2Department of Biotechnology and Biosciences, University of Milano-Bicocca, 20126 Milan, Italy; s.garbujo@campus.unimib.it (S.G.); g.rolla1@campus.unimib.it (G.R.); m.giustra2@campus.unimib.it (M.G.); lucia.salvioni@unimib.it (L.S.); miriam.colombo@unimib.it (M.C.); 3Microscopy Facility, University of Milano-Bicocca, 20126 Milan, Italy; tiziano.catelani@unimib.it

**Keywords:** gold nanoparticles, polymeric-coating, toxicity, safe-by-design, zebrafish, FET

## Abstract

In the biomedical field, gold nanoparticles (GNPs) have attracted the attention of the scientific community thanks to their high potential in both diagnostic and therapeutic applications. The extensive use of GNPs led researchers to investigate their toxicity, identifying stability, size, shape, and surface charge as key properties determining their impact on biological systems, with possible strategies defined to reduce it according to a Safe-by-Design (SbD) approach. The purpose of the present work was to analyze the toxicity of GNPs of various sizes and with different coating polymers on the developing vertebrate model, zebrafish. In particular, increasing concentrations (from 0.001 to 1 nM) of 6 or 15 nm poly-(isobutylene-alt-maleic anhydride)-*graft*-dodecyl polymer (PMA)- or polyethylene glycol (PEG)-coated GNPs were tested on zebrafish embryos using the fish embryo test (FET). While GNP@PMA did not exert significant toxicity on zebrafish embryos, GNP@PEG induced a significant inhibition of embryo viability, a delay of hatching (with the smaller size NPs), and a higher incidence of malformations, in terms of tail morphology and eye development. Transmission electron microscope analysis evidenced that the more negatively charged GNP@PMA was sequestered by the positive charges of chorion proteins, with a consequent reduction in the amount of NPs able to reach the developing embryo and exert toxicological activity. The mild toxic response observed on embryos directly exposed to GNP@PMA suggest that these NPs are promising in terms of SbD development of gold-based biomedical nanodevices. On the other hand, the almost neutral GNP@PEG, which did not interact with the chorion surface and was free to cross chorion pores, significantly impacted the developing zebrafish. The present study raises concerns about the safety of PEGylated gold nanoparticles and contributes to the debated issue of the free use of this nanotool in medicine and nano-biotechnologies.

## 1. Introduction

In recent years, the Safe-by-Design (SbD) approach has gained increasing importance for the development of new nanomaterials, including nanoformulated medicines [1]. According to SbD, the biological hazard of the new nanomaterial needs to be determined at the earlier stage of the production process, in order to predict its potential adverse effect on human health and the environment. Starting from the knowledge of the nanoproduct toxicity, it is, therefore, possible to redesign it and reduce the undesirable effects [2].

Among the many nanodevices developed for biomedical applications, gold nanoparticles (GNPs) have received great interest over the last 10 years for the management of cancer disease, due to their ability in diagnostic and/or therapeutic applications. Indeed, magnetic GNPs are particularly promising for both cancer diagnosis (i.e., by magnetic resonance imaging, X-ray computed tomography, Raman and photoacoustic imaging) and therapy (i.e., plasmonic photothermal and photodynamic therapies, or acting as a drug delivery system) [3].

In the context of an increasingly widespread development of GNPs for oncology, the issue of their biosafety is still today controversial, and many in vitro and in vivo toxicity studies have been performed over the past decade to elucidate the adverse impacts of gold-based nanoparticles [4,5]. Surface properties and stability in biological fluids have been demonstrated to determine GNP toxicity [6,7]. Similarly, shape and size play a key role in the adverse impact of these nanomaterials [7,8] and, in a range between 1 and 15 nm, small-dimension NPs seem to exert more toxic effects than higher-dimension ones [9].

Different strategies have been investigated with the aim of reducing GNP toxicity, including coating with organic stabilizers. Surface organic coating varies the physicochemical properties of the NPs, optimizing their stability and interaction with the target cells, but also making them more biocompatible. Several coatings have been demonstrated effective in reducing the toxicity of inorganic NPs [10,11,12]. Regarding gold-based NPs, in 2014, Cai et al. demonstrated the high biocompatibility of Au NPs coated with amino acids having amphiphilic groups on their surfaces [13]. Among polymers, collagen has proven to be a very efficient coating for gold NP biosafety, even better than dodecylamine-modified poly(isobutylene-alt-maleic anhydride) (PMA) [14]. Nevertheless, PMA coating is still considered one of the most diffused and performant surface modifications for the design of inorganic NPs stable in biological fluids [15,16]. Nanoparticle PEGylation (coating with polyethylene glycol) is also considered a standard strategy to improve NP stability and stealth properties [15]. Systemic toxicity also seems to be prevented by PEG coating [17,18], and several studies demonstrated a good in vitro biocompatibility of PEGylated gold NPs [19,20], especially when compared to nanoparticles engineered with other coatings (i.e., polystyrene sulfonate) [21]. Conversely, some in vivo experiments showed the occurrence of toxicity under exposure to gold NPs coated with PEG5000, including acute inflammation and apoptosis in the liver of mice exposed to 13 nm PEGylated GNPs [22], or an increase in the number of blood cells, as well as liver and kidney damage, upon treatment with 10 and 60 nm PEG-coated GNPs [23]. Coating gold NPs with known stealth polymers, such as PEG, provides other important advantages for biomedical applications, including an improvement in blood circulation, by reducing NP uptake by mononuclear phagocyte systems [24]. Generally, all NPs coated with hydrophilic polymers [25], such as neutral/zwitterionic nanoparticles, are characterized by longer blood circulation, while charged nanoparticles are known to possesses relatively short half-lives [26]. This should also be the same for PMA-coated NPs, whose negative charge [14] is likely responsible for a relatively short half-life in systemic circulation, even though the pharmacokinetics of these formulations has never been described in literature.

However, the issue of the safe use of polymeric-coated gold nanoparticles in biomedical applications is still open, and our study aimed to contribute in this direction by studying the toxicity of polymer-coated GNPs on the vertebrate developmental zebrafish model (*Danio rerio*) using the fish embryo test (FET). Embryotoxicity studies are usually carried out for biomedical applications on developing mammals and, in this context, the use of zebrafish embryos is useful for a prescreening of molecule activity before moving onto investigations on mammals [27]. Moreover, zebrafish is currently the leading alternative to mice for preclinical studies with nanoscale drugs [28], and it has been widely used to evaluate the toxicity of nanoparticles, both at embryonal and at adult stage [29,30], with the benefit of reducing time and cost of mammalian models, in agreement with the 3Rs concept [31]. The toxicity of GNP on zebrafish embryos has also been largely reported in the literature. In 2011, Harper et al. used the zebrafish model to demonstrate how the mortality and malformations of the embryonic stages are dependent on the size and charge of Au NPs. Nanoparticles with no charge are safe toward zebrafish embryos over a broad range of sizes, while AuNPs with both positive and negative charges induce significant adverse impacts on the biological systems [32].

Another study demonstrated that cationic functionalized trimethylammoniumethanethiol (TMAT) AuNPs induce a dose-dependent alteration of eye development and pigmentation, which continues to neuronal deficits and abnormal behavioral activity [33].

So far, the protective activity of different polymeric shells against Au NPs toxicity has not been investigated on developing vertebrates. In the present study, we assessed, for the first time, the toxicity of increasing concentrations of 6 or 15 nm PMA- or PEG-coated GNPs on zebrafish embryos, recording FET endpoints daily up to 96 h after fertilization. The FET represents one of the most widely used standardized protocols, according to Organization for Economic Co-operation and Development (OECD) test guideline n. 236, for the study of the toxicity of compounds and nanomaterials on living organisms, and it is a valid throughput screening for the SbD development of biomedical nanodevices.

## 2. Materials and Methods

### 2.1. Chemicals and Materials

Hydrogen tetrachloroaurate hydrate, 99.999%, (trace metal basis) Acros Organics™ was purchased from Fisher Scientific Italia (Rodano, Italy). Meo-PEG-SH (2000 Da) was purchased from RappPolymere GmbH (Tuebingen, Germany). Boric acid, CHCl_3_, dodecylamine (99%), dodecanthiol, HCl, MeOH, NaCl, NaOH, pentane, poly(isobutylene-alt-maleic anhydride) (molecular weight (Mw) ~6000), sodium borohydride (≥99%), sodium citrate dihydrate (99%), tetrahydrofuran, tetra-*n*-octylammonium bromide (98%), and toluene were purchased from Sigma Aldrich (Saint Louis, MO, USA).

Water used in all procedures was purified by passing through a MilliQ Millipore system.

### 2.2. Synthesis of Spherical Gold Nanoparticles

First, 6 nm spherical gold nanoparticles (6 nm GNPs) were obtained following the Brust–Schiffrin protocol. In a separating funnel, 80 mL of tetra-*n*-octylammonium bromide (49.6 mM) in toluene and 25 mL of HAuCl_4_ (35.3 mM) in MilliQ water were mixed. The solution was shaken for 5 min, the aqueous phase was discarded, and the organic phase was transferred to a round-bottom flask. Then, 25 mL of NaBH_4_ (353.2 mM) in MilliQ water was added dropwise to the organic solution under vigorous stirring. The solution was kept under stirring for 1 h at room temperature (RT) and transferred into a separating funnel. The resulting nanoparticles suspension was washed with 25 mL of HCl (10 mM), 25 mL of NaOH (10 mM), and 25 mL of MilliQ water four times, discarding the aqueous phase after each step. The GNP suspension was kept under stirring overnight; afterward, 10 mL of pure dodecanethiol was added, and the mixture was heated at 65 °C for 3 h under stirring. The sample was centrifuged at 2000 relative centrifugal force (RCF) for 10 min, and the aggregates were removed by discarding the pellet. Then, 250 mL of MeOH was added, and the sample was centrifuged at 2000 RCF, discarding the supernatant. The pellet was dried and dissolved in CHCl_3_, and the obtained GNPs were characterized by means of ultraviolet/visible light (UV/Vis) spectroscopy (NanoDrop^®^ 2000c Spectrophotometer—Thermo Fisher Scientific, Waltham, MA, USA) and dynamic light scattering (DLS) (Malvern Instruments, Malvern, UK). Then, 15 nm spherical gold nanoparticles (15 nm GNPs) were obtained in MilliQ water through the addition of 1.2 mL of HAuCl_4_ (25 mM) into 150 mL of a boiling solution of sodium citrate (1.32 mM). After 15 min, the sample was cooled down and characterized by means of UV/Vis spectroscopy and DLS. Phase transfer in CHCl_3_ was performed by firstly adding MeO-PEG-SH (2000 Da) to the GNP suspension (molar ratio 1:30,000) and keeping the suspension overnight under stirring. Lastly, 50 mL of dodecylamine 0.4 M in chloroform was added to the aqueous GNP suspension. The sample was characterized by means of UV/Vis spectroscopy and DLS [34].

### 2.3. Ligand Exchange of GNP Using MeO-PEG-SH (2000 Da)

First, 6 nm spherical gold nanoparticles were PEGylated by mixing 500 µL of GNPs (566 nM) in CHCl_3_ and 183.8 µL of MeO-PEG-SH (15.4 mM) in CHCl_3_. After 4 h at 50 °C, the solution was dried and dissolved in MilliQ water. The sample was characterized using UV/Vis and DLS. Then, 15 nm spherical gold nanoparticles were PEGylated by mixing 600 µL of GNPs (28 nM) in CHCl_3_ and 10.9 µL of MeO-PEG-SH (15.4 mM) in CHCl_3_. After 4 h at 50 °C, the solution was dried and dissolved in MilliQ water. The sample was characterized using UV/Vis and dynamic light scattering.

### 2.4. Polymer Coating of GNPs

#### 2.4.1. PMA-*g*-Dodecyl Polymer Synthesis

Poly-(isobutylene-alt-maleic anhydride)-*graft*-dodecyl polymer (PMA) was synthetized following a previously published protocol [35] with some modifications. In detail, 3.084 g (20 mmol expressed as in terms of monomer units) of poly(isobutylene-alt-maleic anhydride) (average molecular weight (Mw) ~6000 g/mol) was dissolved in 100 mL of anhydrous tetrahydrofuran (THF). The solution was poured into a 250 mL round-bottom flask containing 2.70 g (15 mmol) of dodecylamine. After mixing (10 min at 800 rpm), the mixture was sonicated for ca. 20 s and heated to 60 °C for 3 h under stirring. Then, the solution was concentrated to 30–40 mL by evaporation of THF under reduced pressure in a rotary evaporator and heated under reflux overnight. The solvent was completely evaporated in a rotary evaporator, and the dried polymer powder was dissolved in 40 mL of anhydrous chloroform, yielding a solution (0.5 M) in monomer concentration.

#### 2.4.2. PMA Coating of GNPs

First, 6 nm and 15 nm GNPs suspended in chloroform were transferred to an aqueous solution by wrapping the amphiphilic polymer PMA around their surface, as described in a previously published protocol [35]. An aliquot of polymer (Vp) at a concentration of Cp = 0.5 M was added to 6 nm and 15 nm GNPs suspended in chloroform. The amount of polymer per NP scales with the effective surface area *A_eff_* of one single NP and with the total number of NPs. The calculation details are shown below; in the case of spherical NPs, *A_eff_* is given as the surface of a sphere, where *d_c_* is the core diameter determined by transmission electron microscopy.
$$A_{eff}=4\pi {\left(\frac{d_{c}}{2}\right)}^{2}=d_{c}\;\pi^{2}.$$

In a solution with volume *V_NP_* and NP concentration *C_NP_*, the total effective surface area of NPs (*A_total_eff_*) can be calculated as follows, where *N_A_* is Avogadro’s number:$$A_{total\_eff}=C_{NP}\times V_{NP}\times N_{A}\times A_{eff}.$$

The number of monomer units that needs to be added per nm^2^ of effective surface area (*R_p/area_*) is a value that should be determined experimentally (in this experiment, *R_p/area_* = 100 nm^−2^ and 3000 nm^−2^ for 6 nm and 15 nm GNPs, respectively). It follows that the number of polymer monomers *N_P_* that needs to be added to NPs suspension is
$$N_{P}=R_{p/area}\times A_{total_{eff}}.\;$$

Consequently, for a polymer stock solution of monomer concentration *Cp*, the volume *V_P_* to be added to the NPs suspension can be determined using the following equation:$$V_{P}=\frac{N_{P}/N_{A}}{Cp}=\frac{R_{p/area}\times A_{total\_eff}}{N_{A}\times Cp}=\frac{R_{p/area}\times C_{NP}\times V_{NP}\times A_{eff}}{Cp}\;$$

The polymer amounts calculated for each nanoparticle population were added into two 250 mL round-bottom flasks containing 6 nm and 15 nm GNPs, and the mixtures were manually stirred for 5 min at room temperature (RT). Then, the chloroform was completely evaporated in a rotary evaporator under heating at 45 °C. A few milliliters of anhydrous chloroform was added to the flasks to reconstitute the solid film, and the solvent was again removed under reduced pressure. These steps were repeated three times to obtain a homogeneous coating. After the last step, the remaining solid film in the flasks was reconstituted in alkaline sodium borate buffer (SBB 30 mM, pH 12 adjusted with NaOH). Then, the polymer-coated GNPs were concentrated using centrifugal filters (Merk Millipore Amicon, 100 kDa), and the polymer excess was removed by means of ultracentrifugation (OptimaXE, Beckman Coulter Life science ultracentrifuge, 100,000× *g*; 1 h at 4 °C). The supernatant was discarded, and the pellet was resuspended in 30 mM SBB, pH 8. This step was repeated three times, and then polymer-coated GNPs were characterized.

### 2.5. Characterization of GNP Suspensions

Uncoated gold nanoparticles were analyzed by transmission electron microscopy (TEM) in order to perform morphological and size characterization. TEM samples were prepared by drying a 5 µL drop of nanoparticle suspension on a carbon film-coated copper grid. TEM micrographs were collected by means of a Jeol JEM 2100Plus (Jeol, Tokyo, Japan) electron microscope, operating with an acceleration voltage of 200 kV and equipped with a 9 MP complementary metal oxide superconductor (CMOS) Gatan Rio9 digital camera (Gatan, Inc., Pleasanton, CA, USA). Plots of size distribution were derived from the analysis of TEM images (three images for each NPs dimension) by the software OriginPro 2019 (OriginLab Corporation, Northampton, MA, USA).

UV/Vis spectroscopy and DLS were used to characterize the hydrodynamic behavior of the GNP@PEG and GNP@PMA suspensions dissolved in MilliQ water and 30 mM SBB, pH 8, respectively. In particular, 10–20 nM of GNP@PEG and GNP@PMA (6 and 15 nm) were analyzed using a Malvern Zetasizer (Malvern Instruments, Malvern, UK), immediately after their preparation. Both the hydrodynamic dimension and the superficial charge of the nanocolloids were expressed as the mean ± SD. UV/Vis spectroscopy was used to calculate the concentration of GNPs. The absorbance at 450 nm was detected and the concentration was calculated assuming a molecular extinction coefficient of 1.26 × 10^7^ M^−1^ cm^−1^ and 2.18 × 10^8^ for 6 nm and 15 nm GNPs, respectively.

To determine the stability of GNP@PEG and GNP@PMA in embryo solution during the FET experiment, NP dispersions was analyzed by UV/Vis spectroscopy analysis immediately after their dilution (T = 0) and after 96 h. The 15 nm GNP@PMA and GNP@PEG were diluted to 1 nM in embryo solution (see Section 2.6.1), previously filtered using a 0.22 µm polyethersulfone (PES) filter, in order to remove precipitates, and analyzed using a NanoDrop^®^ 2000c Spectrophotometer (Thermo Fisher Scientific, Waltham, MA, USA). Then, 500 µL of nanoparticle suspensions were transferred to a 24-well plate and placed in a thermostatic chamber at 26 °C, under static conditions, to reproduce the experimental condition of the FET assay (see Section 2.6.2). After 96 h incubation, nanoparticle suspensions were again analyzed by UV/Vis spectroscopy.

### 2.6. Zebrafish Embryo Treatments and FET

#### 2.6.1. Fish Husbandry and Egg Collection

The adult AB wildtype strain was purchased from the European Zebrafish Resource Center (Karlsruhe Institute of Technology, Eggenstein-Leopoldshafen, Germany). Fish were maintained and bred at the University of Milan-Bicocca zebrafish facility (approved by ATS MetroMilano Prot. n. 0020984—12 February 2018), in a recirculating ZebTec Active Blue aquatic system (Tecniplast, Buguggiate, Italy).

Breeders were kept separated by sexes in 3.5 L tanks at 28 °C, pH 7.5, and 500 µS, under a 14 h/10 h light/dark cycle. Fish were fed three times a day with Zebrafeed (Sparos Lda, Olhão, Portugal).

The day before breeding, adult pairs were located in breeding tanks and separated by a barrier during the night. On the following morning, the barrier was taken out and adults were allowed to mate. Eggs were collected in a strainer within 30 min of mating, rinsed in embryo solution (0.1 g/L instant ocean, 0.1 g/L sodium bicarbonate and 0.19 g/L calcium sulfate), and selected under a stereomicroscope (Zeiss, Oberkochen, Germany).

#### 2.6.2. Fish Embryo Acute Toxicity (FET) Test

Fertilized, undamaged eggs with no malformation, 1 h post fertilization (hpf) and within 3 hpf, were randomly selected from different mates (*n* = 5) and were distributed in 24-well plates. Embryos (20 for each experimental condition) were exposed to 500 μL of embryo solution (control group) or GNP suspensions in a thermostatic chamber at 26 ± 0.5 °C, under static conditions. Control and treated embryos were observed at 24, 48, 72, and 96 hpf. Lethal endpoints (coagulation, lack of somite formation, lack of detachment of the tail from the yolk sac, lack of heartbeat) indicating acute toxicity were monitored every 24 h. Moreover, sublethal endpoints (such as edemas, and tail and eye malformations) and the hatching rate were analyzed. Embryos were observed, and phenotype modifications were acquired through a M205FA stereomicroscope (Leica Microsystems Srl, Buccinasco, Italy) equipped with the LAS-X Expert software.

For the experiments on dechorionated embryos, the chorion was mechanically removed at 24 hpf, and the developing animals were exposed to control solution or GNP suspensions. FET phenotypic observations were performed at 48, 72, and 96 hpf, as reported above.

Median lethal concentration (LC_50_) and hatching time (HT_50_) were calculated using IBM-SPSS Probit Analysis. 

### 2.7. TEM on Embryos

Zebrafish embryos were also analyzed by means of TEM in order to investigate the fate and the biological interactions of Au nanoparticles at the nanoscale. First, 48 h embryos treated with 6 nm and 18 nm Au NPs were fixed for 2 h in fixative solution (2% glutaraldehyde, 2% paraformaldehyde) prepared in phosphate buffer (PB) 0.1 M at room temperature. Subsequently, samples were post-fixed in 1% osmium tetroxide solution prepared in the same buffer and stained overnight in 1% uranyl acetate aqueous solution at 4 °C. After dehydration in a series of alcohol concentrations, samples were infiltrated with graded propylene oxide/epoxy resin solution for 1 day and then embedded in epoxy resin (Epon Epoxy 812). After hardening in the oven at 65 °C for 2 days, samples were cut by means of a Richert-Jung UltracutE ultramicrotome in 70 nm slices. TEM observations and micrograph acquisition were performed as described above (see Section 2.5).

## 3. Results and Discussion

### 3.1. Synthesis and Characterization of GNP@PEG and GNP@PMA

Gold nanoparticles were synthesized by exploiting two different methods, one in organic phase (6 nm GNPs) following Brust–Schriffin’s protocol [36] and the other one (15 nm GNPs) in water phase [34]. 

The resulting 6 nm GNPs were coated with dodecanethiol, a hydrophobic surfactant with high affinity for the gold surface, capable of protecting the NPs from aggregation. Moreover, the presence of alkyl chains allows them to be soluble in an organic solution as chloroform. In contrast, 15 nm GNP were coated with trisodium citrate, used as a reducing agent and surface stabilizer in aqueous solution. In this case, ligand exchange, mediated by MeO-PEG-SH (2000 Da), was necessary to transfer GNPs in chloroform in the presence of dodecylamine (Figure 1).

The two nanoparticles, 6 and 15 nm, were characterized in terms of their core size by TEM (Figure 2A,B), and a plot of their size distribution was obtained by analyzing the TEM images (Figure 2C,D).

Subsequently, the cores were transferred to the water phase by adding PEG or PMA. PEG is a polymer known to interact with metal cores giving special properties [37]. PMA is an amphiphilic polymer obtained by reacting poly(isobutylene-alt-maleic anhydride) with dodecylamine as a hydrophobic side chain. This alkylamine is able to intercalate with the alkylthiol on the GNP surface through a hydrophobic interaction. In basic water, each available succinic anhydride of PMA chains is hydrolyzed to two carboxylates, providing a negative charge to the NP surface and playing a role in colloidal stability by electrostatic repulsion. The NPs were characterized by UV/Vis, to determine GNP concentration, and by DLS, to obtain the mean hydrodynamic diameter (Table 1 and Appendix A).

As Table 1 shows, the difference in surface charge between PMA and PEG coatings was critical, playing a role in colloidal stability and biological properties. In particular, the more negative Z-potential of GNP@PMA could be responsible for the higher interaction of these nanoparticles with the positively charged proteins of chorion [38].

### 3.2. Effect of GNP@PEG and GNP@PMA on Zebrafish Embryos

The toxicity of GNP@PEG and GNP@PMA was evaluated on zebrafish embryonal development, using the FET assay, which allows determining the phenotypic changes of the developing vertebrate upon exposure to the substance tested. The impact of 6 and 15 nm gold nanoparticles coated with PMA or PEG was recorded after a single exposure of the embryos to the NPs at the concentrations of 0.001, 0.01, 0.1, and 1 nM in the maintenance solution, from 3 up to 96 hpf. NP concentrations were chosen in line with the literature [39,40].

The stability of GNP@PEG and GNP@PMA in embryo solution was verified by UV/Vis spectroscopy analysis immediately after their dilution (T = 0) and after 96 h, in the same experimental conditions as the FET assay (in 500 µL, in a 24-well plate put in a thermostatic chamber at 26 ± 0.5 °C, under static conditions). We were able to assess the maximal experimental concentration (1 nM) for 15 nm NPs, while it was not possible to analyze the 6 nm NPs, which were undetectable by UV/Vis at the same dose, even at a 10-fold higher dose. The UV/Vis analysis (Appendix A) did not show significant differences in the absorption spectra of 15 nm GNP@PMA and GNP@PEG at 0 h and 96 h after incubation. The overall shape of the spectra remained the same, with no broadening, and the surface plasmon resonance did not show red-shift, usually ascribable to particle destabilization. Moreover, the absence of a secondary peak appearance at longer wavelengths, typical of aggregate formation, indicates a good nanoparticle stability in embryos solution after 96 h of incubation.

Although all the endpoints predicted by FET were recorded according to the standard protocol, the changes induced by the NPs only concerned embryonic viability, hatching, and tail and eye malformations. The results of these three parameters are, therefore, reported in the present work. 

#### 3.2.1. Embryo Viability

The effect of GNP@PEG and GNP@PMA on zebrafish embryo viability is reported in Figure 3. It was demonstrated that both 6 and 15 nm GNP@PMA did not affect embryo viability, while GNP@PEG induced a decrease in the percentage of living embryos, statistically significant with respect to the control at doses higher than 0.1 nM or 1 nM with 6 nm or 15 nm NPs, respectively. However, a dose-response effect of GNP@PEG on embryo viability seemed to occur only up to 0.1 nM for both NP sizes. On the basis of the zebrafish lethality for treatments with GNP@PEG, LC_50_ values calculated for the two experimental groups were 1.53 and 0.32 nM, respectively. Thus, higher doses of the greater-dimension GNP@PEG NPs were proven to be the most toxic on zebrafish embryo viability.

Despite the instability of GNP@PEG NPs, due to the fast equilibrium of monodentate ligands around the metal surface as supported from the literature [41,42], and although the release of PEG 2000 in aqueous solution upon dilution is feasible, we are confident that the released polymer should not be responsible itself for embryo lethality. In this regard, in 2017, Pelka et al. evaluated the toxic effects of free PEGs of different sizes on zebrafish embryos. They demonstrated that PEG 2000 is by far the least toxic, with an LC_50_ greater than 100,000 ppm [43].

Most of the mortality induced by treatment with GNP@PEG occurred within the first 24 hpf for all tested concentrations, an index of a direct strong impact of these NPs on the first and most susceptible stages of development. The further reduction in viability observed with the higher concentrations over the following hours post fertilization may be related to the incurrence of important morphological alterations, which finally result in embryo death. 

#### 3.2.2. Embryo Hatching

Embryo hatching from chorion is another important observation of the FET, which commonly begins at 72 hpf. Within 96 hpf, more than 80% of embryos hatched from the chorion. Table 2 shows a dose-dependent tendency in stimulating embryo hatching in the presence of GNP@PMA, although no significant difference with respect to the control was observed with both NP sizes, even at higher doses. Hatching anticipation can be associated with a high concentration of PMA-coated NPs on the surface of the chorion, clearly detected using TEM (Figure 4). At 96 hpf, approximately 100% of the embryos escaped the chorion in all experimental groups (Table 2).

Conversely, a strong inhibition of hatching was observed at 72 hpf with 6 nm GNP@PEG starting from the lower doses, and even no hatched embryos were recorded upon exposure to doses of NPs higher than 0.01 nM. No significant effects occurred with 15 nm NPs. The inhibition of hatching was also observed at 96 hpf in embryos exposed to 6 nm GNP@PEG, but this was only significant with 0.01 nM NPs (Table 3). The HT_50_ values calculated on embryos treated with the different concentrations of 6 or 15 nm GNP@PMA or GNP@PEG confirmed a hatching delay with respect to the control in the animals exposed to 6 nm GNP@PEG, with a maximal effect at the dose of 0.01 nM (Appendix A). Therefore, unlike that observed for embryo viability, a significant impact on hatching rate was obtained only with the lower-dimension NPs at a dose far from the maximal tested. These data indicate that the two endpoints, embryo viability and hatching, respond independently to the PEGylated GNPs, and the mortality observed after embryo hatching (Figure 3) does not likely correlate with their ability to hatch from the chorion.

The inhibition of hatching rate observed only with the 6 nm GNP@PEG was not so clear. It could be linked to a higher penetration of these smaller NPs through the embryo skin, which is not favored by the more negative surface charge (Table 1) [44]. It is certainly the first time that an effect of gold-based NPs on zebrafish hatching has been described. In 2011, Asharani et al. performed an FET on embryos exposed to different metallic NPs, including Au NPs. They observed that, while both silver and platinum NPs induced hatching delays, no differences with respect to the control were detected with gold NPs [45]. The hatching delay induced by the small-dimension GNP@PEG could be due to the direct impairment of the NPs on the hatching gland or could be a secondary effect of a motor alteration related to morphological abnormalities, which do not seem to include tail malformations (see Section 3.2.3).

#### 3.2.3. Embryo Malformation

In addition to viability and hatching, other endpoints must be considered according to the FET, including lack of somite formation, lack of heartbeat, non-detachment of the tail, tail malformation (scoliosis and lordosis), and eye defects. Among all these endpoints, abnormal tail flexure and eye defects were identified as the main malformations, only in response to GNP@PEG. Indeed, 6 and 15 nm GNP@PMA did not induce any morphological alteration of embryos at 96 hpf at all tested concentrations except with the lower ones of the smaller size NPs, where a mild lordosis was observed; the percentage malformation was 5.91% ± 0.49% (mean ± standard error (SE), *n* = 20) and 10.74 ± 3.26 (mean ± SE, *n* = 20) for 0.001 and 0.01 nM, respectively (Figure 5A). Conversely, GNP@PEG exerted a strong impact on embryo morphology (Figure 5B), inducing tail malformation (lordosis, scoliosis) with 0.001–0.1 nM 6 nm NPs and 0.01–0.1 nM 15 nm NPs. Eye defects were induced only with 6 nm GNP@PEG and, in particular, monolateral microphthalmia was frequently observed in the embryos exposed to these NPs. As reported above, almost half of the embryos exposed to both sizes of 1 nM GNP@PEG died under the effect of the NPs. Nevertheless, no malformations were detected in the surviving embryos. The percentage malformation, reported in Table 4, does not indicate a large number of malformed embryos in the groups which responded to the treatments with 6 nm NPs, whether in terms of tail or eye defects, while up to ~45% tail malformations were observed in those exposed to 15 nm GNP@PEG. Although the low and non-dose-dependent incidence of tail malformations in the embryos exposed to 6 nm GNP@PEG does not correlate with the embryo viability data, the maximal toxic impact on tail morphology obtained with 0.1 nM 15 nm NPs is in line with the viability results. We cannot, therefore, exclude a certain correlation between the mortality recorded after 24 hpf at this dose of 15 nm GNP@PEG (Figure 3D) and the impairment of tail development. For sure, the effect of GNP@PEG on tail morphology does not correlate with the hatching behavior of embryos considering that the higher tail malformation incidence was recorded upon exposure to 15 nm NPs, which were completely inactive on the hatching index (Table 3).

The high toxicity of GNP@PEG on zebrafish embryos seems to not agree with our previous studies on other developing vertebrates, where PEG was able to prevent the mortality and malformations of *Xenopus laevis* induced by CuO and ZnO NPs [12]. Nevertheless, GNP@PEG is almost neutral and its dimensions are much smaller than those of PEG-MeOs NPs (hydrodynamic size = 1800–3000 nm, Z-potential = 11–14 mV), and the developmental pattern of the two animal models differs significantly, e.g., in the characteristic of the enveloping membranes, hatching time, and timing of stomodeum perforation. These findings suggest that the surface PEGylation, coupled with the NP size, metal core, and possible dissolution of toxic ions, may represent P-chemical factors able to modulate the toxicity response in a species-specific fashion, depending on the targeted biological systems. For sure, the small size of GNPs allowed an easier tissue penetration, determining a stronger toxic effect not prevented by the polymeric coating.

Nevertheless, our data agree with previous in vitro results describing that the cytotoxic impact of PEGylated gold NPs is higher than that of non-PEGylated NPs. In particular, even though PEGylation significantly reduced nanoparticle uptake, at a similar number of internalized GNPs, a stronger intrinsic cytotoxic effect was observed with PEG-coated ones, due to the induction of oxidative stress. The reason for this is still unclear; however, PEGylated NPs are not well tolerated by cells [46]. In any case, the toxic effects on zebrafish embryos are attributable to the impact of PEGylated NPs and not of the released PEG, which has been demonstrated to be totally safe on different cell lines [47], such as on zebrafish [43], in the form of 2 kDa chains.

### 3.3. Interaction of GNP@PEG and GNP@PMA on Zebrafish Embryo Chorion

To verify if the lack of toxicity of GNP@PMA on zebrafish embryonal development could be due to the inability of these NPs to cross the chorion and reach the animal during the first stages of the development, we analyzed by TEM the embryos in the chorion exposed for 48 h to the nanoparticles (Figure 4). A large amount of both 6 and 15 nm GNP@PMA was observed on the chorion surface. The 15 nm GNPs appeared highly dispersed, while small nanoparticles had a tendency to produce aggregates. Nevertheless, at higher magnification, a great density of single 6 nm GNP@PMA was also visible embedded into the chorion outer layer. The interaction of these NPs with the chorion seems plausible due to the net negative charge provided by PMA to the nanoparticles (Table 1), which would, therefore, interact with the positive charges of the proteins constituting the chorion [38]. Even though a great number of NPs seemed to be sequestrated by chorion external layers, no passage through the pores was detectable. Therefore, GNP@PMA is mostly impaired in reaching the embryos before hatching, and this can explain the lack of effect on embryo survival (Figure 3) and tail morphology (Figure 5A) with these NPs. 

To verify if GNP@PMA sequestration by the chorion is indeed responsible for their inability to overcome this barrier and, therefore, exert a significant toxicity on the inner embryos, we evaluated the effect of 6 and 15 nm NPs after mechanical dechorionation (at 24 hpf). No relevant effects on embryo viability (percentage of living embryos at 48–96 hpf vs. living embryos at 24 h) were observed with 6 nm GNP@PMA. On the contrary, a certain impact on zebrafish viability was obtained with 15 nm NPs, which induced a slight increase in embryo lethality at 0.01 nM (Appendix A). No malformations were detected in the embryos exposed to both sizes of NPs. Thus, even after a direct exposure of the embryos to GNP@PMA, the adverse effects induced by these NPs were not so relevant. It is important to highlight that, since dechorionation is a quite invasive technique that can be exploited only starting from 24 hpf, the embryos are exposed to the treatment for a shorter period of time with respect to the standard FET assay (3 days vs. 4 days); moreover, they are not exposed to the NPs on the first day of their development, on which they should be more sensitive to the treatment. However, we can conclude that the lack of penetration of the GNP@PMA through the chorion, mainly due to its capture on the surface proteins, is only in part responsible for the safety of these NPs in zebrafish embryos. Our results are in line with the study of Soenen and coworkers, which demonstrated, using a multiparametric approach, the lack of cytotoxicity of PMA-coated 4 nm Au NPs up to a concentration of 10 nM [48].

Unlike GNP@PMA, GNP@PEG did not adhere massively to the surface of the chorion. Indeed, very little GNP@PEG was detectable outside the chorion, as single (15 nm) or aggregated (6 nm) nanoparticles (Figure 4). It is, therefore, feasible that the PEG-coated NPs, not entrapped by chorion proteins, were more facilitated in reaching the inner embryos to exert their toxic effect. 

Surprisingly, no NPs were detected by TEM in the chorion, neither in proximity of the embryo surface nor in the animal tissues. Nevertheless, we have to take into account that these small and highly dispersed nanoparticles, incubated at very low concentrations (nM), cannot be easily identified by electron microscopy if they do not accumulate on biological surfaces or into compartments. 

## 4. Conclusions

From the data obtained on zebrafish embryos analyzed according to FET, it seems evident that, while different sizes of GNP@PMA do not exert significant toxicity on this developmental vertebrate model, the same NPs functionalized with PEG induce strong adverse effects, more or less evident from different FET endpoints according to their dimension and treatment concentration. In particular, both 6 and 15 nm GNP@PEG were able to induce a significant inhibition of embryo viability at the higher concentrations, starting from 24 hpf. Moreover, inhibition of the hatching rate, obtained under the effect of the smaller GNP@PEG, and the incidence of malformations, in terms of tail morphology and eye development (only with 6 nm NPs), were additional outcomes of embryo exposure to the pegylated NPs. The 6 nm NPs seem to be more effective than 15 nm NPs, taking into account hatching percentage and the occurrence of eye malformation, but a lower percentage of tail malformations was detected with these nanoparticles with respect to the higher-dimension ones. Further investigation is necessary to obtain a better comprehension of the size-dependent toxicity of GNP@PEG on zebrafish embryos, since the signaling involved in the morphogenesis of ectodermal-derived cephalic structures and mesodermal-derived caudal ones may be impacted differently.

In terms of the different toxicological behavior of PMA- and PEG-coated Au NPs on zebrafish embryos, it is reasonable to hypothesize that it could be mainly related to the different net charge of these NPs, which in turn affects their interaction with both the chorion and the embryo surface. Our hypothesis is that, while neutral GNP@PEG does not associate with the positive proteins of chorion surface and freely crosses pores by passive diffusion to interact with the developing zebrafish, GNP@PMA is largely entrapped by the chorion and does not reach the embryo in a sufficient amount to induce significant toxicity. However, only a mild toxic response was observed on the embryos directly exposed to GNP@PMA, after dechorionation. In this sense, it was recently demonstrated that the internalization of gold NPs into zebrafish skin is also affected by their surface net charge, and that neutral Au NPs are able to penetrate deep into the skin cells through a disruption of the membrane bilayer, while anionic gold NPs did not get into the bilayer and remained adsorbed on the surface [44]. 

The overall results obtained on the developmental vertebrate model zebrafish, put in evidence that PMA is promising in terms of safety for the design of gold nanotools. Nevertheless, further toxicological evaluations on different experimental models are mandatory to confirm the potential of PMA coating to prevent Au NP toxicity. On the other hand, our data support the idea that PEG coating is not such a good candidate for the SbD development of Au-based biomedical nanoparticles. In addition, the longer blood half-life expected for the neutral GNP@PEG is likely responsible for prolonged side-effects on the treated organism, especially when compared to the negative and more clearable GNP@PMA.

Therefore, although the importance of NPs PEGylation is commonly accepted to improve the efficiency of therapeutic delivery by reducing systemic toxicity [18], our study raises concern about the use of this polymer as a coating for the prevention of Au NP toxicity in adult and developing organisms. Accordingly, the present study provides an important contribution to the debate on the in vivo safety of PEG-coated colloidal NPs.

## Figures and Tables

**Figure 1 nanomaterials-11-01004-f001:**
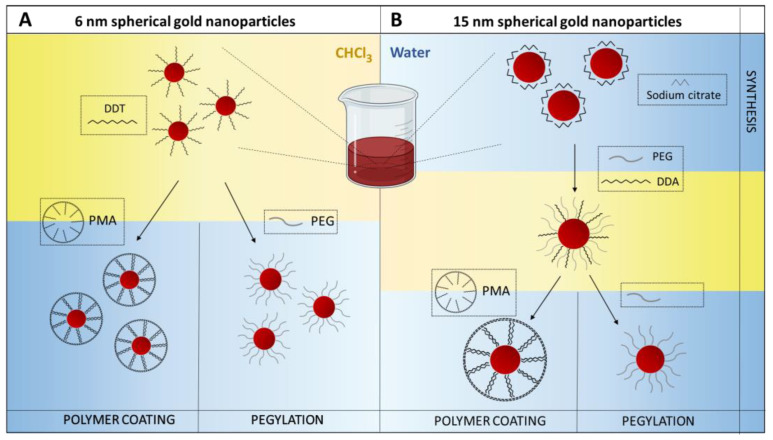
Steps for obtaining 6 nm (**A**) and 15 nm gold nanoparticles (GNPs) (**B**) coated with poly-(isobutylene-alt-maleic anhydride)-*graft*-dodecyl polymer (PMA) or polyethylene glycol (PEG). (**A)** 6 nm GNPs functionalized with dodecanethiol (DDT) were synthesized in organic phase and then transferred to an aqueous solution by coating with the amphiphilic polymer PMA or functionalization with PEG molecules. (**B**) 15 nm GNPs were obtained according to a two-step method: following the synthesis in citrate solution, 15 nm GNPs were transferred to organic solvent via the addition of a mixture of PEG and dodecylamine (DDA), where PEG drove the aqueous-to-organic phase transfer. Then, similarly to 6 nm GNPs, 15 nm GNPs were transferred back to an aqueous solution by coating with the amphiphilic polymer PMA or functionalization with PEG.

**Figure 2 nanomaterials-11-01004-f002:**
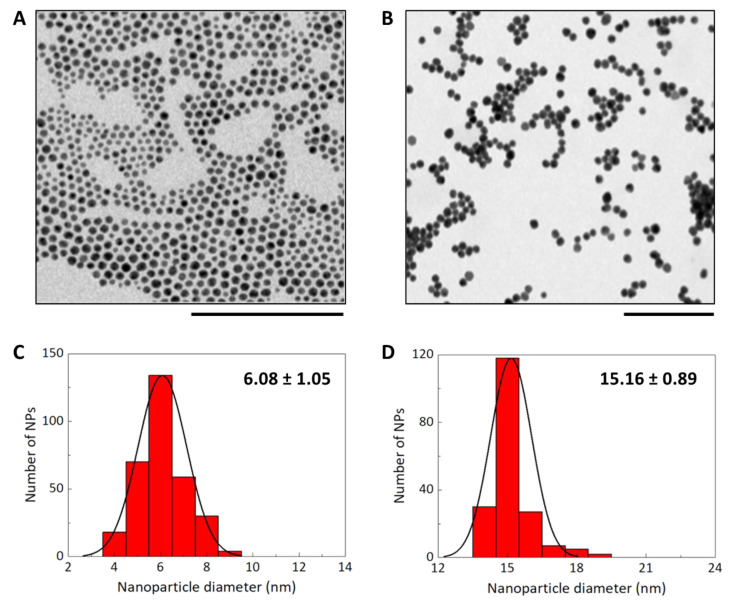
Transmission electron microscopy (TEM) images of GNP 6 nm (**A**) and 15 nm (**B**). Bars: 100 nm for (**A**) and 200 nm for (**B**). From the analysis of TEM images, plots of size distribution were derived for 6 nm (**C**) and 15 nm (**D**) NPs. Mean ± SD of 315 or 189 measurements for 6 or 15 nm NPs, respectively.

**Figure 3 nanomaterials-11-01004-f003:**
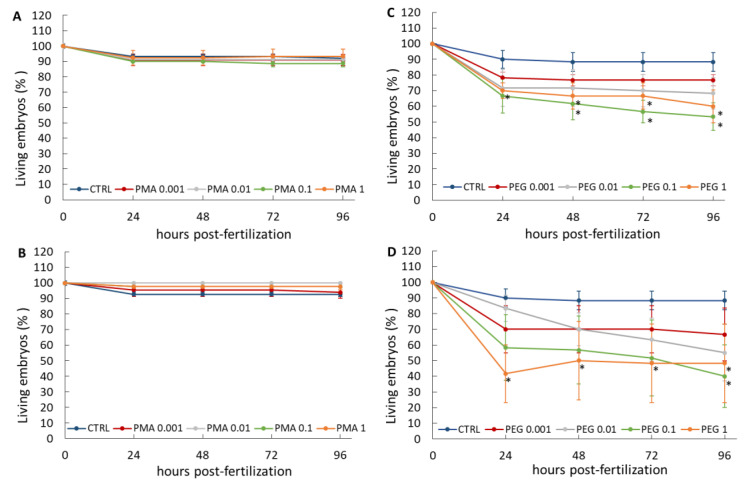
Viability (%) of zebrafish embryos exposed or not (control (CTRL)) to 0.001–1 nM GNP@PMA 6 nm (**A**) or 15 nm (**B**), or GNP@PEG 6 nm (**C**) or 15 nm (**D**). Mean ± standard error (SE) of three different experiments. * *p* < 0.05 vs. the corresponding CTRL (one-tailed *t*-Student test).

**Figure 4 nanomaterials-11-01004-f004:**
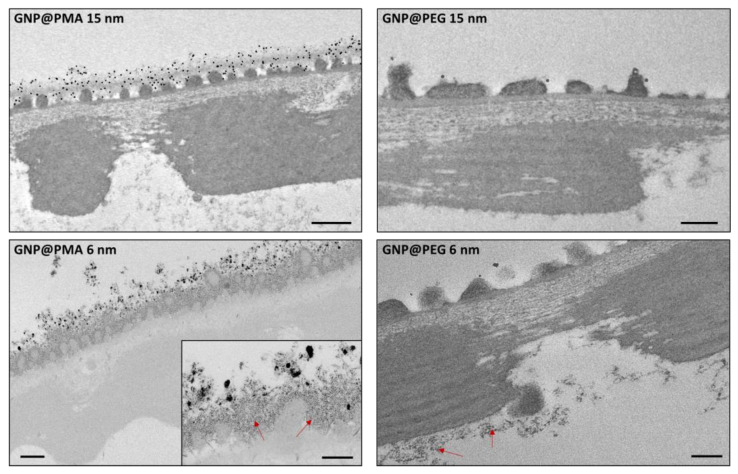
TEM images of zebrafish embryos (48 hpf) incubated with 6 nm or 15 nm GNP@PMA or GNP@PEG, at a concentration of 1 nM. The high-magnification insert highlights a great amount of dispersed 6 nm GNP@PMA on the chorion surface (red arrows); blue arrows indicate rare 15 nm GNP@PEG or 6 nm GNP@PEG aggregates on the chorion surface. Bars = 500 nm in the figures of GNP@PMA and 200 nm in the figures of GNP@PEG and in the insert of 6 nm GNP@PMA.

**Figure 5 nanomaterials-11-01004-f005:**
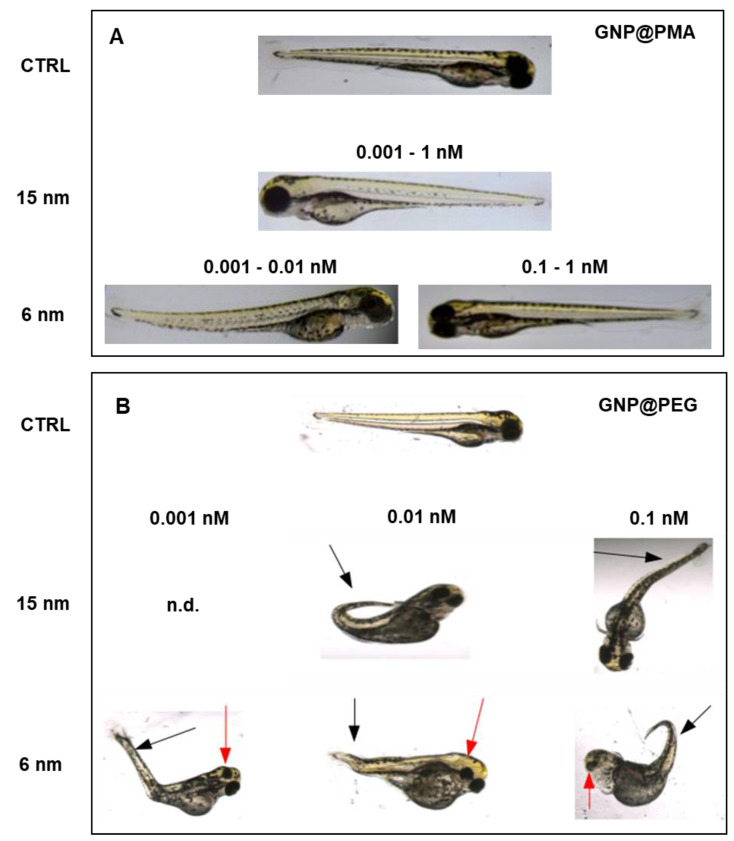
Tail and eye malformations of zebrafish embryos exposed or not (CTRL) to 0.001–1 nM GNP@PMA (**A**) or GNP@PEG (**B**), 6 nm or 15 nm; n.d., not detected.

**Table 1 nanomaterials-11-01004-t001:** Characterization of GNP@PEG and GNP@PMA by dynamic light scattering (DLS). Z-potential, hydrodynamic size (distribution in number), and polydispersity index (PDI) as measured by DLS. Mean ± SD of three independent experiments.

Coating	PMA		PEG	
**Size**	6 nm	15 nm	6 nm	15 nm
**Z-potential (mV)**	−25.7 ± 1.17	−12.5 ± 0.65	−0.79 ± 0.09	−0.09 ± 0.09
**Hydrodynamic size (nm)**	12.4 ± 0.37	22.0 ± 1.52	17.5 ± 0.55	24.9 ± 0.81
**PDI**	0.567 ± 0.013	0.462 ± 0.004	0.182 ± 0.014	0.297 ± 0.013

**Table 2 nanomaterials-11-01004-t002:** Hatching percentage of embryos exposed to 0.001–1 nM 6 or 15 nm GNP@PMA. Mean ± SE of 3–7 experiments.

hpf	Ctrl	0.001 nM		0.01 nM		0.1 nM		1 nM	
		6 nm	15 nm	6 nm	15 nm	6 nm	15 nm	6 nm	15 nm
**0–48**	0	0	0	0	0	0	0	0	0
**72**	13.69 ± 6.96 (7)	13.99 ± 4.80 (4)	27.56 ± 18.73 (3)	28.98 ± 11.51 (4)	32.22 ± 28.95 (3)	22.09 ± 9.44 (4)	34.05 ± 30.54 (3)	35.67 ± 10.24 (4)	45.95 ± 27.47 (3)
**96**	100 (7)	98.21 ± 1.79 (4)	100 (3)	100 (4)	100 (3)	100 (4)	100 (3)	100 (4)	n.d.

**Table 3 nanomaterials-11-01004-t003:** Hatching percentage of embryos exposed to 0.001–1 nM of 6 or 15 nm GNP@PEG. Mean ± SE of three experiments.

hpf	Ctrl	0.001 nM		0.01 nM		0.1 nM		1 nM	
		6 nm	15 nm	6 nm	15 nm	6 nm	15 nm	6 nm	15 nm
**0–48**	0	0	0	0	0	0	0	0	0
**72**	26.76 ± 4.50 (3)	6.55 ± 0.30 ** (3)	22.12 ± 3.94 (3)	0	38.05 ± 3.32 (3)	0	21.11 ± 10.60 (3)	0	21.43 ± 17.50 (3)
**96**	98.04 ± 1.96 (3)	89.58 ± 10.42 (3)	89.56 ± 6.45 (3)	58.40 ± 15.15 * (3)	90.74 ± 4.90 (3)	82.50 ± 11.81 (3)	91.67 ± 8.33 (3)	73.29 ± 13.39 (3)	83.33 ± 13.61 (3)

** *p* < 0.01; * *p* < 0.05 vs. CTRL (*t*-Student test).

**Table 4 nanomaterials-11-01004-t004:** Percentage malformation of embryos exposed to 0.001–1 nM of 6 or 15 nm GNP@PEG. Mean ± SE of three experiments.

Malformation %	Ctrl	0.001 nM		0.01 nM		0.1 nM		1 nM	
		6 nm	15 nm	6 nm	15 nm	6 nm	15 nm	6 nm	15 nm
**tail**	0	8.63 ± 5.46 (3)	0	10.45 ± 6.18 (3)	37.41 ± 21.59 (3)	9.70 ± 4.99 (3)	45.00 ± 18.93 (3)	0	0
**eye**	0	11.61 ± 2.68 (3)	0	5.71 ± 2.97 (3)	0	7.20 ± 3.73 (3)	0	0	0

## Data Availability

No supporting data to the present study are available.

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
