# Peer review of "The Role of Polymeric Coatings for a Safe-by-Design Development of Biomedical Gold Nanoparticles Assessed in Zebrafish Embryo"

_nanomaterials, 2021, doi:10.3390/nano11041004_

Round 1

Author Response

attached file

Reviewer 2 Report

The study suggests GNP@PMA is safe in zebrafish embryos, which can be a promising biomedical nanodevices; while NGP@PEG inhibits embryos viability, delays hatching, increases incidence of malformations.

The study is clear and the data can support the conclusion.

Points:

The characteristics and advantages of PMA coated gold nanoparticles have been studied and reported, what is the novelty of this study.

Since GNP-PEG per se have adverse effect on viability, how can the data differentiate the GNP-PEG effect on hatching and malformation when the treatment is affecting the viability?

Please clarify why those concentrations of NP were tested? And is the toxic effect a combination of the dose and the NP size?

Will the authors use other models for the study such as mice that are more commonly used for the study of toxicity? Please discuss.

Author Response

attached file

Reviewer 3 Report

The authors have assessed the toxicity of two different coated gold nanoparticles with two different sizes on zebrafish embryos. They found clear differences of toxicity for both coatings, being the polyethylene glycol (PEG) coat much more deleterious than the poly(isobutylene-alt-maleic anhydride) (PMA) coat. The authors conclude that PMA coated gold nanoparticles might potentially be safer than the PEG gold nanoparticles for biomedicine applications.

The results of both nanoparticles on zebrafish embryos are clear and I barely have objections against these results as such. However, I have several points that should be clarified by the authors before publication.

The reported effects are based on effects on a widely used biological model typically considered for embryotoxicity studies. Thus, the results found in this manuscript should be initially addressed to embryotoxicity and the results should be taking into consideration for biomedical applications on pregnant women. Please, clarify this point.

The authors have offered a plausible explanation about the differences in embryotoxicity for both coating. However, this explanation is based on the existence of a barrier (chorion) that does not exists in humans. Thus, I wonder whether PMA gold nanoparticles are indeed safer than PGE gold nanoparticles in humans.

The physical characterization of the nanoparticles is poor. At least, the plot with the distribution of sizes should be added in order to determine the proportion of the total population of nanoparticles with sizes of 6 and 15 nm.

What about the stability of nanoparticles in embryo solution? How the authors can demonstrate that during 96 hours of exposure the nanoparticles have not aggregated and/or precipitated generating a new substance with different physical properties? In this line, the differences between 6 and 15 PEG nanoparticles (Table 3), could be attributed to differences in bioavailability?

MINOR POINTS

It is necessary to use abbreviations in a more consistent way. By example, in abstract there are non-defined abbreviations as PEG and PMA. To define an abbreviation for a single use (by example SbD) seems to be inappropriate.

I suggest preparing Tables 2, 3 and 4 in a different format. It is very difficult, especially in Table 2, to differentiate the hatching percentage for 72 hours among different columns.

Section 4 (Conclusions) is too long and this is because most of the text is a mere repetition of the points already presented and discussed in section 3. Thus, this section can be notably reduced.

Author Response

attached file

Round 2

Author Response

-Lines 81-88: The sentence in line 86 of the Introduction “It should be 86 the same for PMAcoated NPs too” needs to be rewritten and clarified. GNP@PMA nanoparticles reported in this manuscript are negatively charged (lines 307, 537, 591, 29). Then, according to line 86, PMAcoated NPs are expected to possess relatively short half-lives.

 We have rewritten the sentence according to Reviewer suggestion (Lines 87-88)

 - Due to the relevance of the results achieved with GNP@PMA that are reported in the manuscript, with lower toxicity than GNP@PEG, the authors should include a statement about this issue again in the Conclusions. Thus, in lines 614-619 the authors comment on the concerns in using PEG-coated GNP due to their toxicity, but they should also mention that, however, longer half-lives are expected for the neutral GNP@PEG. Thus, surface modification of NPs with PEG has been used as a universal therapeutic technique in the literature to prepare long-circulating NPs, as PEGylation is thought to prevent or decrease uptake by the reticuloendothelial system.

 We have added also in Conclusion a statement about the importance of the surface charge of GNP coated with the two different polymers on their blood circulation, and the consequent systemic toxicity (lines 615-618) .

-"Figure S3: % of living embryos at 48, 72 and 96 hpf, after dechorionation and exposure for to 6 nm or 15 nm GNP@PMA." Not all data shown in the legend are displayed in Figure S3, I guess because the measured embryos’ viability values for the 6 nm GNP@PMA test are all close to 100% (as mentioned in line 546). If that is the case, please indicate it in the graph or in the figure caption. Also, please revise typos in the legend of Figure S3 (PMA should be written in capital letters).

 We have revised the Caption of Figure S3 according to reviewer’s suggestion. PMA in legend has been also revised.

-Line 479: the sentence “gold NPs are not positively charged “ should be replaced by “GNP@PEG are almost neutral“.

We have revised the sentence as indicated by the reviewer.

Reviewer 3 Report

The authors have successfully addressed all my previous concerns. Thus, I have no further objections and suggest publication of this manuscript.

Author Response

We are pleased to have fully satisfied the requests of the reviewer